# Activity-Based Restorative Therapy Promotes Progression from Asymmetry to Symmetry in Posture and Gait in a Child with Chronic, Incomplete Spinal Cord Injury

**DOI:** 10.3390/children10030594

**Published:** 2023-03-20

**Authors:** Laura Leon Machado, Kathryn Lucas, Andrea L. Behrman

**Affiliations:** 1Frazier Rehabilitation Institute, UofL Health, Louisville, KY 40202, USA; 2Department of Neurosurgery, University of Louisville, Louisville, KY 40202, USA; 3Kentucky Spinal Cord Injury Research Center, Louisville, KY 40202, USA; 4Kosair Charities Center for Pediatric NeuroRecovery, Louisville, KY 40202, USA

**Keywords:** non-traumatic spinal cord injury, pediatrics, incomplete spinal cord injury, neurorecovery, asymmetry, gait, rehabilitation

## Abstract

Incomplete spinal cord injuries (ISCI) in pediatrics and adults can lead to asymmetric motor impairments exhibiting as asymmetries of posture and gait. Recently, rehabilitation guidelines for adults with neurologic injuries have focused on gaining a functional gait pattern as measured by speed and distance, even if asymmetry deficits persist. Activity-based restorative therapies (ABRT) take advantage of activity-dependent neuroplasticity to change an individual’s neuromuscular capacity. This is a report of an ambulatory child with chronic ISCI presenting with significant postural and gait asymmetries who enrolled in an ABRT program. Across 79 ABRT sessions, the child gained symmetry during sitting, standing, and walking. Even though this child was a functional ambulator at enrollment, targeting symmetry of movements via improved neuromuscular capacity further enhanced her achievement of kinematically appropriate function for participation in daily activities.

## 1. Introduction

Pediatric and adult incomplete spinal cord injury (ISCI) can result in asymmetric patterns of paralysis, paresis, and abnormal sensation [1,2,3]. For the pediatric population, inherently undergoing musculoskeletal growth, repetitive use of asymmetric motor patterns uniquely subjects children to the risk of secondary musculoskeletal complications (e.g., scoliosis, joint laxity, joint instability) [4,5]. Ambulation and posture, though visibly asymmetrical due to sensorimotor asymmetries such as muscle imbalances and weakness and balance and/or coordination deficits, may be deemed “functional” because the individual can effectively participate in daily activities and ambulate [6,7]. Walking patterns in both children and adults after ISCI, however, exhibit deficits beyond decreased speed and endurance, including altered foot placement, lack of reciprocal stepping coordination, variations in joint motions throughout the step cycle, decreased time in the swing phase, asymmetric (increased, decreased, or abducted) arm swings, decreased self-selected gait speed, use of assistive devices, and increased double support time [6,8,9,10,11,12,13]. As noted in case reports, these gait deviations negatively impact quality of life [14], increase oxygen consumption [14], and increase metabolic cost [15]. While such asymmetric, compensated walking patterns in children may be labeled as “functional,” many children are still at risk for tripping, slipping [16], or worsening musculoskeletal dysfunctions [4]. With time, these conditions can restrict the distances or places that children walk.

This case highlights the transition of a functionally independent ambulatory child with chronic ISCI from a prevalence of asymmetric posture and gait kinematics to more symmetrical posture and locomotor patterns through enrollment and completion of a clinical Activity-Based Restorative Therapy (ABRT) program. The aim of ABRT is to activate the neuromuscular system above, across, and below the lesion after spinal cord injury to improve the neuromuscular capacity, health, and quality of life of individuals with SCI. To this end, ABRT applies biological principles of activity-dependent plasticity whereby provision of appropriate sensorimotor input through repetitive, task-specific, intense practice activates the neuromuscular system sub-serving daily activities. The principles of ABRT emphasize kinematically-consistent limb movements and postures, maximizing weight-bearing through the legs, task-specific sensory input, and minimizing the use of compensatory strategies, whether behavioral or physical assistance [17,18,19,20]. Guided by clinical reasoning and the premises that for a growing child, symmetry in posture and movements (i.e., sitting, standing, sit-to-stand movement, walking) is superior to compensated repetitive asymmetry and that quality of movement matters to the quality of life and participation in the present and in the future, we sought to assess if ABRT could change postural and gait pattern asymmetry to relative symmetry while maintaining function and participation in school, home, and community.

We thus selected Activity-Based Restorative Therapy as the primary clinical intervention, guided by principles of activity-dependent neuroplasticity, to promote neuromuscular activation and control below the level of the SCI and for the child to achieve the use of appropriate, task-specific kinematics during a typical day. [21] In this case, the clinical goals were set as achieving a sustainable alignment of typical trunk upright posture and upper and lower limb kinematics for daily activities of stability and mobility [17,18,19,20].

## 2. Case Presentation

The child’s parents provided informed consent for the child’s clinical data. Intervention history and outcome measures were stored in an approved database and used for program evaluation. The study was conducted in accordance with the Declaration of Helsinki and approved by the Institutional Review Board of the University of Louisville (IRB# 05.016J). All analyses were conducted clinically with video.

### 2.1. Medical History from Chart Review

At 4 years, 2 months of age, this previously healthy child presented with a stomachache and left hip pain. These complaints were followed by a rapid onset of symptoms (24 h), progressing to urinary incontinence, constipation, leg pain (left > right), and bilateral lower extremity flaccid paralysis leading to an inability to stand, walk, or bear weight through her legs (Figure 1). She could not sit independently due to trunk weakness. She had no respiratory complaints. Sensory changes were present across the legs and perianal areas. Initial medical interventions were guided by a suspected diagnosis of transverse myelitis. MRI showed a well-circumscribed lesion on the left aspect of the spinal cord at T9 demonstrating a heterogeneous signal on T2-weighted sequences, as well as a second lesion at the level of T11/12 centrally within the spinal cord. Upon repeat MRI 2 weeks later, both lesions were still noted but had decreased in size. The running diagnosis was changed to cavernous malformation leading to spinal cord infarct. Additional neuroimaging, performed 2 months later, showed a linear streak of blood in the center canal from T6 to T12, with residual cavernous malformation still present at T9. The following additional diagnoses were ruled out: acute disseminating encephalomyelitis, transverse myelitis, acute flaccid myelitis, Zika virus, neuromyelitis optica, Guillain-Barre’s syndrome, and West Nile virus.

The child remained in acute care for 3 weeks followed by 1 month of inpatient rehabilitation. Upon discharge, she was able to advance her legs during ambulation but needed moderate support for balance. She continued with outpatient physical therapy 2x/week and occupational therapy 1x/week. Medical equipment prescriptions included solid ankle-foot orthoses (AFOs) for the left leg, night stretching braces, a gait trainer, a walker, and a manual wheelchair. Botox was recommended but not performed for the left gastrocnemius, should the left heel continue to have difficulty staying in the AFO. At 5 months post-injury, the physician reported that right lower extremity function (from a motor standpoint) was back to normal but that there was continued left-sided weakness and increased tone. Medical reports document changes in sensation, from initial decreased sensation at bilateral lower extremities to diminished pain and temperature sensation in the right leg (5 months post-injury) and emerging sensation resulting in complaints of neuropathic pain in the left leg. Gabapentin was prescribed to assist with this discomfort.

Although this child did not have a formal diagnosis of Brown Sequard syndrome [2,3], her clinical presentation (motor and sensory) showed similarities to this type of injury.

### 2.2. Initiation of Activity-Based Restorative Therapy

Nine months post-ISCI, the child enrolled in an outpatient ABRT program at 4 years, 11 months old (Figure 1). While the child was able to sit and stand independently, she demonstrated postural asymmetries.

### 2.3. Postural Alignment

Sitting: Her preferred postural alignment was left trunk rotation with a right lateral shift, for which she compensated by leaning left at the thorax to maintain her head upright, creating a postural scoliotic position. She maintained her weight shifted towards the right hip (Figure 2B). Initially, the child’s Segmental Assessment of Trunk Control (SATCo) score was 11/20, meaning that when sitting 90/90 at the hips and knees with feet on the floor, she had the ability to maintain appropriate alignment above the support provided at the lower ribs and pelvis to achieve a neutral trunk and pelvic alignment at and below the level of support [22].

Standing: When standing, she maintained the same trunk postural alignment observed while sitting with the addition of a narrow base of support and a left lateral weight shift, with her left knee hyperextended and her right knee flexed (Figure 2B). Her standing pediatric reach test was 5.0 cm forward, 2.0 cm to the right, and 3 cm to the left [23] (Figure 2A).

### 2.4. Gait

She demonstrated an asymmetric gait pattern, dominated by a right lateral shift of the trunk with a left trunk flexion at the rib cage to achieve upright alignment, decreased knee flexion through the stance phase on her right leg, and hyperextension of the left knee noted during the stance phase, resulting in veering, a hip drop, and vaulting. Her right arm was held in abduction to the trunk, and her left arm swing excursion was minimal and used as a counterbalance tool to aid in balance (see Appendix A). She was an ambulator in her home and preschool environments without an assistive device but utilized a stroller for community mobility needs.

### 2.5. Intervention

The ABRT program emphasized recovery of postural and gait symmetry and control with appropriate kinematics at a high intensity 5x/week for 1.5 h/day. This schedule allowed her to continue attendance in her half-day preschool. In therapy, for the first hour of the session, the child stood and stepped with manual facilitation by the therapist and trainers to achieve appropriate alignment in a body weight-supported treadmill environment, followed by a half-hour off the treadmill in an overground environment [17,20,24]. Emphasis was on symmetry of movements during tasks such as stepping, stairs, standing, and reaching with appropriate kinematics and sensory input (Figure 3). Re-assessments were completed every 20 sessions, approximately 1x/month. The following measures from the ABRT program’s standardized outcome measure bank were completed (see Figure 2A for scores): 10 m walk test [25,26], 2 min walk test [27], SATCo [22,28], Pediatric Reach Test [29], Pediatric Balance Scale [30], and Spinal Cord Injury Functional Ambulation Inventory (SCI-FAI) [31].

### 2.6. Follow-Up

Follow-up evaluation was performed after discharge to track the child’s functional independence and mobility symmetry. During follow-up evaluation planning, the family updated her medical history. This family reported that the child had undergone a spinal detethering surgery since discharge.

## 3. Results

The child completed 79 ABRT sessions across a 4-month episode of care, with a high compliance rate of 94% attendance. As the child demonstrated changes in her neuromuscular capacity, the daily session focus changed to continually challenging the neuromuscular system, promoting symmetry and control during sitting, standing, and walking (Figure 3). The first 20 sessions (~1 month) focused on symmetry of movements spatially and temporally, with manual facilitation provided to the trunk, pelvis, and both lower extremities by the therapist and trainers. The trunk position was assisted so it was upright, with pelvis rotation and translation promoting equal weight shifts. The steps were narrow but not crossing midline on the treadmill, and coordination of the trunk, pelvis, and leg supports reinforced a cyclic sinusoidal center-of-mass transfer between legs. The legs were manually facilitated to minimize hyperextension at the (left) knee and vaulting on the right leg and increased flexion through the swing on the left (Figure 3). This emphasis reduced the tendency for the right leg to step in the center of the treadmill and vault while the left circumducted and hyperextended. Symmetry and equal weightbearing were reinforced in overground environments, including an upright trunk and posture, reinforcing equal weight-bearing during all standing and squatting tasks (Figure 3). We instructed her to use a posterior rolling walker at home and in the community to decrease her overground walking speed and help her maintain appropriate gait (limb kinematics) and trunk symmetry. While the child was an independent walker, changing her environment by slowing her down, adding a posterior walker, and promoting symmetry of gait helped her effectively practice the new spatial and temporal patterns at home and in the community.

During the next 20 sessions, the trainers facilitated the movement of the left leg and pelvis during treadmill-based stepping while assessing the patient’s ability to maintain kinematics of the right leg independently, demonstrating changes in her neurological capacity to transition to independent performance of gait mechanics (Figure 3). In overground environments, the focus shifted to training and practice of gait kinematics without facilitation as the patient improved independent control of trunk, pelvis, and limb kinematics (Figure 3). During the last 20 sessions, manual facilitation was required only to maintain appropriate kinematics of the child’s left leg while the child independently controlled the right leg with appropriate kinematics (Figure 3). Overground, symmetric eccentric control was emphasized through activities such as squats and hops (Figure 3). The patient transitioned from ambulation with a posterior rolling walker to ambulation with two hiking poles for dissociation of arms and legs and for promotion of arm swing with upright posture/trunk symmetry. This was a preparatory step to transition to no assistive device upon discharge (Figure 3).

### 3.1. Postural

Sitting: The patient’s trunk alignment improved from the initial evaluation to discharge (see Figure 2B for outcome measures).

Standing: Her standing posture progressed to an upright trunk alignment, with equal weight-bearing and bilateral knees extended by discharge (Figure 2B). She demonstrated improvements in functional reach (see Figure 2B) and by discharge exhibited forward reach within normative data for her age [32]. Additionally, her pediatric balance scale increased from 46/56 to 51/56 [30].

### 3.2. Gait

Her SCI-FAI [31] gait parameter score changed from 14/20 (weight shift absent on the right, stance foot obstructing the swing foot on the left, foot placement obstructing the swing limb on the right, toe drag at the initiation of the swing phase on the left, and lack of heel contact before the forefoot bilaterally) to 19/20 (Figure 4A), with the left foot lacking heel contact before the forefoot contact. Gait speed was maintained throughout the intervention during the 10-m walk (1.45 m/s pre-intervention and 1.22 m/s post-intervention), which is within or faster than the self-selected gait speed of uninjured peers (1.11 ± 0.12 m/s). During the 2-min walk test, her gait speed did not change (0.97 m/s pre-intervention and 1.12 m/s post-intervention) and remained within the normal range of the self-selected gait speed of uninjured peers. She displayed improvements in arm swing, trunk positioning, and lower extremity alignment, leading to more symmetric kinematics for arm swings and trunk, hips, and knees, as well as ankle/foot positions (Figure 4A). Upon video review, the stance percentage of her gait cycle for the left limb (60%) was within normal limits, with the right (68%) exceeding the normal (Figure 4B), starting above the age-appropriate normative range (58 ± 2%) [33]. Following treatment, the stance cycle percentages for left and right were more normalized by a reduction in the amount of time spent in single limb stance on the right (Figure 4B). Her double-limb support time decreased from 28% pre-intervention to 13% post-intervention, which is within the normative range (16 ± 4%) for her age [33] (Figure 4B).

Follow-up evaluation: A follow-up evaluation was performed five months post-discharge (Figure 2B). At the follow-up evaluation, the family reported that at 1-month post-discharge, there had been increased falls, and the child had subsequently undergone detethering surgery.

## 4. Discussion

This case identifies the opportunity for improved kinematics of posture and gait post-ISCI in a child presenting with asymmetrical weakness/motor involvement and loss of pain and temperature in the contralateral limb using activity-based restorative therapy. Even though this child initially fully participated at home, school, and even play (i.e., walk fast, run) in what could be described as a “functional” walker (see Appendix A) at baseline, she exhibited asymmetry throughout her trunk, pelvis, leg, and arm movements during sitting, standing, and walking. While rehabilitation guidelines for locomotor function following chronic neurologic injury in adults have recently placed an emphasis on participation and achieving functional goals via intensity of training targeting speed and distance goals [34,35], this child’s case supports the use of therapeutic intervention to address asymmetries of gait. Outcomes of improved symmetry in posture and walking facilitate continued age-appropriate participation. Children, adolescents, and adults with incomplete spinal cord injuries have displayed the potential for neurorecovery for the initiation of ambulation and increased neurological capacity through ABRT interventions [17,18,19,20,21,24,36,37]; however, this case exemplifies the possibility for recovery of symmetry of posture and gait in a child who is ambulatory. ABRT [14,21] targets the inherent biological response of activity-dependent plasticity [37] via task-specific, intense practice to activate more typical sensorimotor patterns of muscle activation above, across, and below the lesion. This therapeutic approach activates a more efficient, age-appropriate neuromuscular pattern and decreases habitual maladaptive patterns in gait. Durability is demonstrated through the maintenance of gained symmetric movements up to 5 months after discharge, even post-surgery. There were, however, some changes in her follow-up scores, including decreased balance and gait speed. Several intervening variables may contribute to these changes, including the residual effects of detethering surgery and the ongoing influence of musculoskeletal growth. A child experiencing a growth spurt may benefit from brief, repeat bouts of therapy to assist in responding to musculoskeletal adaptations. Further broadening the utility of this intervention into clinical care may be assisted by review of scientific background and utility [20] as well as participation in continuing education courses that provide standardized training for the therapy team (including both hands-on training and clinical decision making).

Optimizing therapeutic interventions is critical, as a lack of kinematically appropriate gait has the potential to limit children in their motor development and lifelong physical activity [38]. If children with SCI are not efficient in their pattern of mobility, they may be at risk of transitioning to a more restrictive assistive device or wheelchair during adolescence and adulthood [39]. Alternatively, age-appropriate gait kinematics and parameters lend themselves to increasing children’s physical activity, both in childhood and into adulthood, due to improved physical literacy [40,41] and the ability to perform higher cardiovascular-demanding tasks with greater efficiency. Future studies demonstrating changes across biomechanical outcomes (e.g., motion capture or EMG recordings) could add greater sensitivity and clarity to the observational changes reported in this clinical case report. Additionally, advances in other forms of neuromodulation (e.g., transcutaneous stimulation) in adults and pediatrics will be of interest to clinical and research studies that seek to change movement patterns from asymmetry to symmetry [42,43,44].

Daily episodes of ABRT successfully improved from predominantly asymmetrical movements to predominant symmetry in gait, trunk posture, and arm swing without sacrificing function, independence, or participation. The incidence of long-term secondary musculoskeletal complications, e.g., scoliosis, may be reduced with the child’s improved neuromuscular control for daily movement patterns and progression to more typical and symmetrical posture and locomotion. Targeting symmetry of movements via improved neuromuscular capacity may further enhance the goals of function and participation for ambulatory children post-ISCI.

## Figures and Tables

**Figure 1 children-10-00594-f001:**
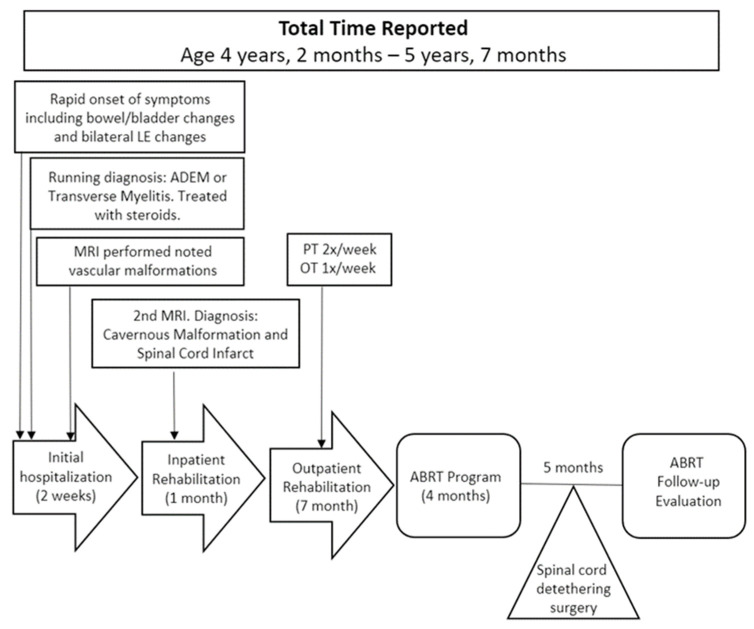
Timeline of medical history from injury (4 years, 2 months) through ABRT follow-up evaluation (5 years, 7 months). LE = lower extremity, ADEM = Acute disseminated encephalomyelitis, MRI = Magnetic resonance imaging, PT = physical therapy, OT = occupational therapy, ABRT = activity-based restorative therapies.

**Figure 2 children-10-00594-f002:**
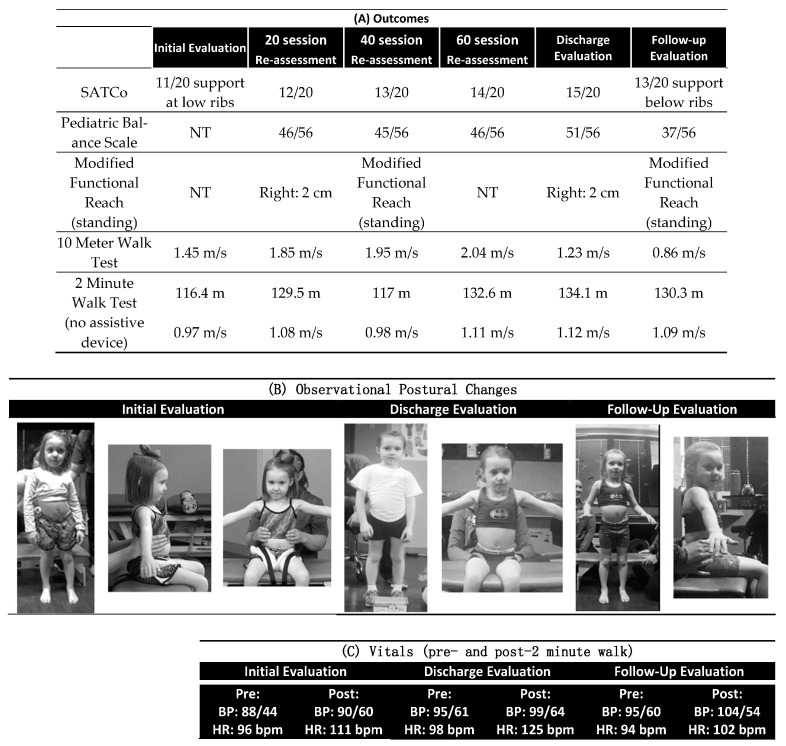
(**A**) Outcome measure scores from 20 session re-assessments (initial evaluation through follow-up evaluation. (**B**) Observational Postural Changes = Improved symmetry of shoulder height, decrease in trunk lateral curve and more equal weight bearing on the legs in standing and through the pelvis in sitting. (**C**) Vitals taken pre- and post-2-minute walk test. NT = This test was added to the program’s standardized measurement bank after this patient’s initial evaluation. SATCo = Segmental Assessment of Trunk Control, m/s = meters/second, BP = blood pressure, HR = heart rate, bpm = beats per minute, cm = centimeters.

**Figure 3 children-10-00594-f003:**
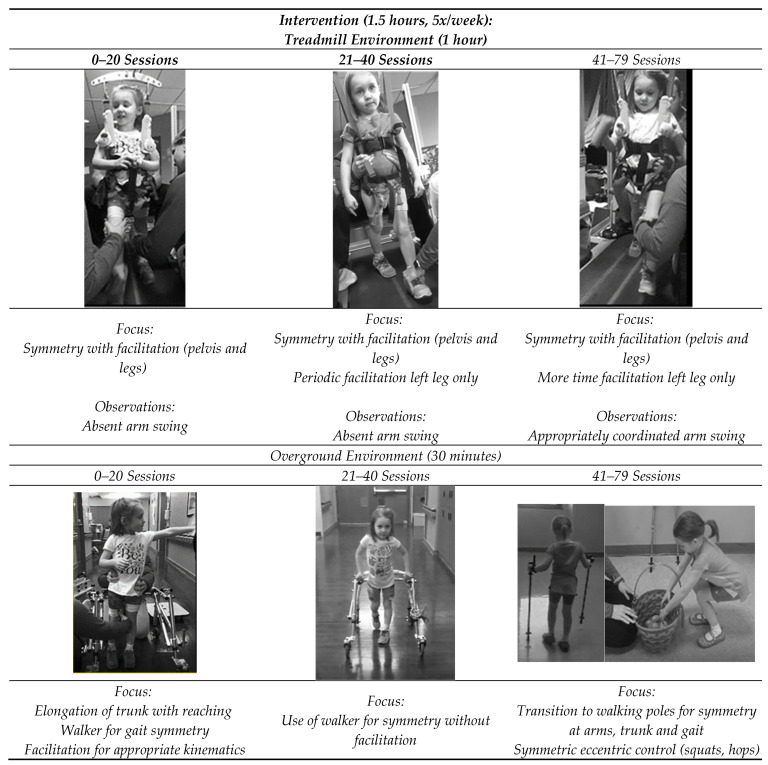
Progression of Activity-Based Restorative Therapy (ABRT) intervention. Focus changed across 3 intermittent periods between re-assessments at 20, 40 and 79 sessions. Trends towards symmetry were noted in arm swing, stepping pattern and trunk.

**Figure 4 children-10-00594-f004:**
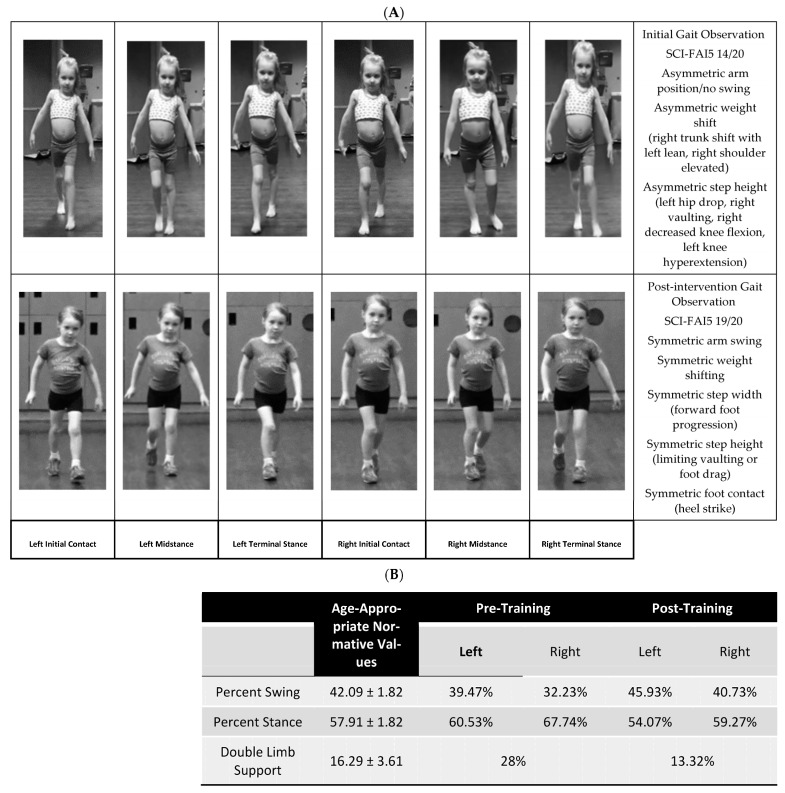
(**A**) Observational change in gait pattern symmetry and objective changes captured by SCI-FAI from initial evaluation through discharge evaluation. SCI-FAI = Spinal Cord Injury Functional Ambulation Index. (**B**) Percentages of swing and stance phases before and after treatment compared to age-appropriate normative values from Voss et al. 2020 [29].

## Data Availability

The data that supports the findings of this study are available upon reasonable request from the corresponding author: Andrea L. Behrman andrea.behrman@louisville.edu.

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
