# Peer review of "Activity-Based Restorative Therapy Promotes Progression from Asymmetry to Symmetry in Posture and Gait in a Child with Chronic, Incomplete Spinal Cord Injury"

_children, 2023, doi:10.3390/children10030594_

Round 1

Reviewer 1 Report

A good case study looking at the impact of activity-based restorative therapy on posture correction in children with chronic incomplete spinal cord injury. It is essential to have data from such case studies because it significantly provides invaluable data about the effectiveness of corrective measures such as the ABRT program presented here. Also, observing and reporting the changes across 79 ABRT sessions is thoughtfully designed.

The introduction must flow well to keep the average reader interested. 

The authors have the necessary information to describe the premise, but it just needs to flow well. 

Line 38: extra space

The timeline in Fig 1 is presented very clearly and concisely 

The quality of Figure 2 should be improved, similar to figure 3, which is clear, although it might be a formatting issue. Please make sure to have a more precise figure.

The process, results, and discussion are well-written and presented. I would encourage the authors to add more discussion on the future possibilities which could improve the conditions and the finding this could be added to the existing therapies to increase efficiency. 

Author Response

Thank you for your time and thoughtful review of this manuscript. We have addressed your points in the revised document (see tracked changes) and figures to add clarity.

Introduction was edited to improve flow and clarity.

Extra space removed from line 38. Figure 2 was modified for clarity and resolution of image improved.

Future possibilities that may influence existing therapies efficiency were added to line 249-254. 

Reviewer 2 Report

In this manuscript, Leon Machado et al. report a case study of a participant with pediatric spinal cord injury performing activity-based restorative therapy to improve gait and posture symmetry. This is an important area of research, as the effects of intense physical therapy interventions in pediatric populations are relatively poorly understood, and may have a great longitudinal impact on independence and quality of life. The manuscript is well-written and the study appears to be well-performed. However, I have some methodological and novelty concerns that need to be addressed prior to consideration of acceptance that are discussed in detail below.

Major Comments

·       It would be helpful to include a definition sentence for Activity-Based Restorative Therapy in the introduction for readers who are less familiar with this approach.

·       The methods section references Figure 2 in the sitting and standing subsections to visually demonstrate the participant’s clinical presentation. Is this referring to the initial evaluation pictures? It is difficult to visualize the described issues and compensation strategies in these pictures. Could the diagrams be labeled to show the transition from initial evaluation to the discharge and follow-up evaluation?

·       The study participant demonstrated a decrease in performance across tasks in her follow-up evaluation compared to her performance at discharge. Do the authors attribute this to a decrease in the quantity of task performance over this time period, the surgery that was performed, or a combination? Would this data suggest that continuous intense rehabilitative protocols should be implemented to maintain gains observed through ABRT, especially in the pediatric population where musculoskeletal growth may cause rapid changes?

·       There have been multiple case studies using intense locomotor training in the pediatric SCI population (including by these authors and referenced in this manuscript). The authors should explicitly emphasize what new information is being gleaned from this participant’s experience compared to previous case studies (e.g., outpatient vs inpatient, solely focused on improving asymmetry, etc.).

·       The introduction describes maintaining function in school, home, and community. Could the authors provide further description of how the study was organized to prevent interference in typical childhood activities? Also, how could this approach be applied in a clinical setting to achieve similar results more broadly?

Minor Comments

·       Neither of the two references for the first sentence of the introduction reference pediatric spinal cord injury, instead mostly focusing on Brown-Sequard syndrome. This should be revised.

·       Line 29: “them” could be made clearer (e.g., patients/participants)

·       Lines 34-39: “Walking patterns in both children and adults…” Why do some of the items in this list have citations and some do not? Either each item should be confirmed with a citation, or there should be grouped list of citations at the end of the sentence.

·       Lines 39-40: “These gait deviations negatively impact quality of life [12], increase oxygen consumption [12],…” These statements are based solely on a case report, and should be confirmed by further literature citations.

·       Line 42: typo “at risks” -> “at risk”

·       ABRT is defined twice in the introduction.

·       Line 64: “Intervention history and outcome measures to be stored in an approved research database…” Why is this sentence in future tense?

·       Line 93: “Botox was recommended for the left gastrocnemius should the left heel continue to have difficulty staying in the AFO.” Was this procedure performed or only recommended? Unclear based on the wording.

·       Lines 137-138: “Reassessments were completed every 20 sessions, approximately 1x/month.” Did this include obtaining the outcome measures shown in Figure 2? If so, it would be interesting to show the longitudinal progression of the participant’s performance.

·       Why wasn’t the Pediatric Balance Scale and Modified Functional Reach performed prior to ABRT and the data is listed as performed at the 20th session evaluation?

·       The walking speed of the 10 meter walk test decreased over time. Do the authors have a hypothesis for why this occurred (e.g., increased independence, focus on symmetric movements, day-by-day variability in measurement)?

·       The detethering surgery should be discussed in the methods section along with details about the follow up-visit.

·       The ASIA exam is understandably difficult to perform in the pediatric population. However, were there any clinically detectable changes observed in motor or sensory function due to the ABRT intervention in addition to the reported measurements?

·       Although extremely difficult to perform in the pediatric population, it would have been interesting to evaluate kinematic or muscle activation changes through motion capture/EMG recordings. This could be listed as a limitation/future direction to further quantify the effects of ABRT.

·       It would also be interesting to discuss future studies on the effects ABRT with and without neuromodulation interventions such as transcutaneous spinal stimulation, which has been shown to improve posture in the pediatric SCI population (Keller et al. 2021, Nature Communications), or epidural stimulation which has been shown to improve gait in the adult SCI population (Gill et al. 2021 Frontiers in Systems Neuroscience).

Author Response

Thank you for your time and thoughtful review of this manuscript. We have addressed your points in the revised document (see tracked changes).

Definition of ABRT included in lines 52-60. 

Time point descriptors (Initial evaluation, discharge evaluation and follow up evaluation) were added to observational posture changes in figure 2. 

Comment on follow up evaluation scores added, see lines 255-260. 

Clarity (additional references) to new information this case report brings to literature added, see lines 245-248.

Additional information on efforts to maintain typical childhood routine, see lines 143-144. 

Descriptions on how to apply this approach to clinical settings added in lines 263-267. 

Added pediatric SCI reference to line 27. 

Line 29: clarified "them" to the children (line 29)

Line 34-39: moved all citations to end of sentence (line 39)

Line 39-40: clarified "as noted in case reports" (line 39)

Line 42: edited typo (line 43)

Removed repeated definition of ABRT (line 67)

Line 64: edited sentence to reflect past tense (line 76)

Line 93: Added clarity to patient's medical history (line 103)

Line 137-138: Added data for all time points of re-assessments (Figure 2A)

Added clarity to why pediatric balance scale and modified functional reach not preformed on initial evaluation (see Figure 2 legend)

Cause of decreased walking speed across time (lines 255-260)

Details of detethering added to Methods (lines 156-160)

We did have additional insight into sensory changes across her episode of clinical care. As indicated, an ASIA exam in the pediatric population is difficult and literature questions reliability for children under 10 years.  

Statement on further studies including EMG/motion capture added (lines 275-279)

Statement on further studies including other forms of neuromodulation (lines 278-281).

Round 2

Reviewer 2 Report

The authors have addressed my previous comments, and the manuscript has improved as a result. I congratulate the authors on a well-written manuscript.